# Differential Brain Activation for Four Emotions in VR-2D and VR-3D Modes

**DOI:** 10.3390/brainsci14040326

**Published:** 2024-03-28

**Authors:** Chuanrui Zhang, Lei Su, Shuaicheng Li, Yunfa Fu

**Affiliations:** Faculty of Information Engineering and Automation, Kunming University of Science and Technology, Kunming 650500, China; z970425@hotmail.com (C.Z.); scli_511@hotmail.com (S.L.); fyf@ynu.edu.cn (Y.F.)

**Keywords:** affective brain–computer interface, affective computing, EEG, emotion recognition, virtual reality

## Abstract

Similar to traditional imaging, virtual reality (VR) imagery encompasses nonstereoscopic (VR-2D) and stereoscopic (VR-3D) modes. Currently, Russell’s emotional model has been extensively studied in traditional 2D and VR-3D modes, but there is limited comparative research between VR-2D and VR-3D modes. In this study, we investigate whether Russell’s emotional model exhibits stronger brain activation states in VR-3D mode compared to VR-2D mode. By designing an experiment covering four emotional categories (high arousal–high pleasure (HAHV), high arousal–low pleasure (HALV), low arousal–low pleasure (LALV), and low arousal–high pleasure (LAHV)), EEG signals were collected from 30 healthy undergraduate and graduate students while watching videos in both VR modes. Initially, power spectral density (PSD) computations revealed distinct brain activation patterns in different emotional states across the two modes, with VR-3D videos inducing significantly higher brainwave energy, primarily in the frontal, temporal, and occipital regions. Subsequently, Differential entropy (DE) feature sets, selected via a dual ten-fold cross-validation Support Vector Machine (SVM) classifier, demonstrate satisfactory classification accuracy, particularly superior in the VR-3D mode. The paper subsequently presents a deep learning-based EEG emotion recognition framework, adeptly utilizing the frequency, spatial, and temporal information of EEG data to improve recognition accuracy. The contribution of each individual feature to the prediction probabilities is discussed through machine-learning interpretability based on Shapley values. The study reveals notable differences in brain activation states for identical emotions between the two modes, with VR-3D mode showing more pronounced activation.

## 1. Introduction

Human emotions encompass psychological and physiological manifestations linked to subjective feelings, temperament, personality, motivational inclinations, behavioral reactions, and physiological arousal [1,2]. Emotions influence people’s cognition, decision-making, and interpersonal interactions [3]. Positive emotions may contribute to improving daily work efficiency, while negative emotions may inconvenience people’s normal lives [4].

Recently, numerous researchers have increasingly focused on the domain of brain–computer interfaces. These interfaces utilize neural activity produced by the brain to facilitate seamless interactions between users and computers [5]. Concurrently, the emergence of affective artificial intelligence in human–machine interaction (HMI) has attracted growing interest [6]. The integration of emotions into human–computer interactions has rapidly evolved into a multidisciplinary research domain known as affective computing, encompassing computer science, neuroscience, psychology, and cognitive science [7]. Emotion recognition is one of the issues in the field of affective computing [8], with potential applications in disease assessment, fatigued driving, and psychological workload estimation [9].

In the domain of emotion recognition, methods for detecting affective states can be broadly categorized into two groups: nonphysiological signals and physiological signals. The former includes facial expressions [10], speech [11], and gestures [12], while the latter relies on physiological signals like electroencephalogram (EEG), electrocardiogram (ECG), electromyogram (EMG), galvanic skin response (GSR), and respiration (RSP). Compared to nonphysiological signals, which can be easily feigned in emotion recognition, physiological signals are considered more objective and reliable in conveying emotions [13]. Among these physiological signals, the EEG is recognized for its excellent temporal resolution, enabling direct emotion recognition by analyzing instantaneous brain activities elicited by emotional stimuli [14]. Recently, there has been growing interest in enhancing Brain–computer Interfaces by leveraging user emotional state information obtained from EEG, a concept known as the Affective Brain–computer Interface [15].

Many researchers have conducted emotion recognition experiments on various EEG datasets. These datasets are predominantly created using classic emotional induction paradigms to stimulate and record the resulting brainwave signals, for instance, the popular DEAP [2], SEED [7], and MAHNOB-HCI [16] datasets. However, The classic paradigm of emotion induction relies on passive emotion induction in laboratory settings, which differs significantly from real-world contexts and cannot fully encompass the psychological and physiological components of participants [17]. Therefore, eliciting intense and multifaceted emotions in participants is challenging, hindering the understanding of emotional processes. An ideal emotional induction paradigm should immerse participants in more realistic environments and adjust based on their responses, thereby facilitating deeper engagement in EEG-based emotion research.

Virtual reality (VR) serves as a potential medium to bridge the gap between laboratory and real-world environments. VR offers highly immersive and realistic complex virtual environments. VR also allows for experimental control, fully immersing users in the created surroundings and potentially enhancing their emotional experiences [18]. In virtual environments, emotions can be naturally and authentically evoked [19], offering benefits in entertainment, education, and psychotherapy [20]. Due to its unique features such as immersion and interactivity, VR could be an excellent solution for emotional induction paradigms.

In the application of VR scenarios, VR scene display methods include helmet-based, desktop-based, and projection-based methods, with helmet-based methods being more commonly employed for virtual scene display. Similar to traditional imaging, VR imagery also encompasses two modes: nonstereoscopic (VR-2D) and stereoscopic (VR-3D). Currently, the VR-2D mode is the predominant method for presenting VR films, as its production is relatively straightforward. However, VR-3D live-action films require more expensive filming equipment, a more complex production process, and more stringent broadcasting conditions, but they offer a more intense stereoscopic sensation and more realistic effects than VR-2D films.

Recent researches have compared differences in brain activation during emotion arousal between VR-3D and traditional 2D environments. For example, Yu et al. [21] examined the neural mechanisms underlying two visual experiences, VR-3D and traditional 2D, based on EEG, while subjects viewed positive and negative emotional videos. They discovered that the β and γ networks exhibited higher global efficiency in the VR-3D group. Tian et al. [22] investigated the impact of two visual modes, VR-3D and traditional 2D, on emotion arousal. They found stronger emotional stimuli and greater emotional arousal in the VR-3D environment, along with higher beta EEG power identified in VR-3D than in traditional 2D. Xie et al. [23] conducted an emotion induction experiment involving six basic emotions to record EEG signals while participants watched VR-3D and traditional 2D videos. They found significant differences in induced discrete emotions between the VR-3D and traditional 2D modes, with greater brain activation observed in the VR-3D mode. However, these studies have focused on traditional 2D and VR-3D environments, with less extensive research on the disparities between the VR-2D mode and the VR-3D mode.

Given the constraints of prior investigations, the principal aim of this study is to delineate disparities in brain activation patterns using EEG-based head-mounted VR displays across VR-2D and VR-3D modalities. Additionally, we aim to incorporate more pertinent EEG signal data to enhance the accuracy of EEG emotion recognition. The principal research contributions of this paper can be delineated as follows: (a) selected effective EEG stimulation materials and collected dataset; (b) a hybrid model combining Differential Entropy (DE), Convolutional Neural Network (CNN), and Long Short-Term Memory Network (LSTM), proposed in this paper.

## 2. Methods

To investigate the disparities in EEG signals between the VR-2D mode and the VR-3D mode, this study conducted EEG experiments in both modes. The experimental design comprises six steps. First, EEG data from subjects were collected in VR-2D and VR-3D modes, with their subjective questionnaire scores. Second, the EEG signals were preprocessed to obtain clean and high-quality signals. Third, power spectral density (PSD) and DE features were extracted from the preprocessed EEG signals. Fourth, data analysis was performed. Fifth, feature selection was performed through recursive feature elimination. Sixth, machine learning and deep learning were applied to classify the EEG data. Last, feature importance was discussed based on Shapley-value machine learning interpretability. Figure 1 illustrates the process of VR emotion induction and analysis.

### 2.1. Participants and Ethics

This study recruited 32 undergraduate and graduate students by posting recruitment advertisements on campus, detailing the study’s theme and criteria such as age range and health status. Throughout the recruitment process, strict adherence to ethical and legal standards was maintained, employing rigorous recruitment procedures. Participants were aged between 21 and 26 years (mean age = 24.23 years, standard deviation = 4.15 years), with an equal gender distribution of 16 males and 16 females. All participants were right-handed and randomly assigned to either the VR-2D or VR-3D group, each consisting of 16 individuals (8 males and 8 females). Medical history information was obtained through questionnaire surveys, confirming no history of psychiatric disorders or brain trauma, and normal or corrected-to-normal vision. Due to signal artifacts, two participants were excluded from the final analysis, resulting in data analysis based on 30 participants (15 males and 15 females). All participants provided voluntary consent and received comprehensive information regarding the research objectives, experimental procedures, and associated risks. Prior to commencement, participants signed written informed consent forms and received appropriate compensation upon completion, with the option to withdraw from the experiment at any time without consequences. Ethical approval for this study was obtained from the local ethics committee.

The selection of undergraduate and graduate students as participants was based on several considerations. Firstly, these two groups typically demonstrate higher academic levels and relatively good physical health, which helps minimize external factors’ interference with experimental results. Additionally, they represent a wide age range and diverse educational backgrounds, enhancing the universality and representativeness of the study findings. Finally, undergraduate and graduate students usually possess strong learning and adaptability skills, enabling them to better understand and comply with experimental requirements and procedures.

The sample size of 32 participants was chosen to strike a balance between considerations of research design and resource constraints. This sample size selection aims to balance the statistical power of the experiment with the feasibility of available resources. Although a sample size of 32 participants may not cover all individual differences, such sample sizes are relatively common in similar experiments to achieve significance and reliability of results.

### 2.2. Experimental Equipment

The experimental setup, as illustrated in Figure 2, comprises the following components: a computer equipped with an Intel Core i7 processor(Intel Corporation, Santa Clara, CA, USA), GTX 3060 graphics card(NVIDIA Corporation, Santa Clara, CA, USA), and 16 GB RAM(Micron Technology, Inc., Boise, ID, USA), and an HTC Vive head-mounted (HTC Corporation, New Taipei City, Taiwan) display for presenting VR videos. The inter-pupillary distance of each participant is measured and adjusted accordingly. EEG signals were collected using the Neuracle EEG Recorder (Borui Kang (Changzhou) Co., Ltd., Changzhou, China), which includes a cap with 32 EEG channels. Electrode placement follows the 10/20 system. To ensure EEG signal quality, electrodes are filled with conductive gel, and during the experiment, electrode impedance should be kept below 5 kΩ. REF serves as the reference electrode, while GND is the ground electrode.

### 2.3. Measurement Metrics

This experiment utilized the widely employed Self Assessment Manikin (SAM) for measuring individual emotional responses. SAM, designed by Bradley and Lang [24], is a participant subjective rating scale that directly measures the emotional valence and arousal levels that humans experience in response to different stimuli, including indices such as valence, arousal, and dominance. In this experiment, scoring was focused primarily on valence and arousal, with rating scales ranging from 1 to 9 (the valence scale ranges from unhappy to happy, with higher numbers indicating greater happiness; the arousal scale ranges from calm to excited, with higher numbers indicating stronger excitement).

To ensure the validity and fidelity of the SAM, we conducted an extensive literature review and performed preliminary testing before the experiment. Additionally, detailed usage instructions were provided to participants to ensure their correct understanding and utilization of the scale. Through these measures, we aimed to accurately capture participants’ emotional experiences using SAM in our study.

### 2.4. Stimuli Selection

Given the deficiency in required expertise [25], the inability to access gold-standard equipment, suitable controlled environments, and the VR materials in the field of affective computing [26], the emotional elicitation video stimuli utilized in the experiment were sourced from an immersive VR video public database at Stanford University [27]. This dataset includes 73 immersive VR clips, each assigned a valence and arousal score within the four quadrants of Russell’s emotion model [28]. On this valence–arousal (VA) plane, the four quadrants are HAHV, HALV, LALV, and LAHV.

For the HAHV, LAHV, and LALV quadrants, videos with valence and arousal scores closest to the extreme corners of the quadrants were selected. Considering the absence of videos in the HALV quadrant in the dataset, this experiment adopted emotion-eliciting videos previously utilized in a paper by Li et al. [25], where researchers selected 15 of the most viewed horror videos on YouTube. Each video was rated by at least 16 volunteers according to the discrete 9-point SAM scale for valence and arousal. For each video x, the normalized arousal and valence scores were calculated by dividing the average score by the standard deviation $\mu_{x}/\sigma_{x}$. Ultimately, researchers chose two videos closest to the extreme corners of the VA plane quadrant as HALV videos. The HALV videos used in this paper can be found online at https://www.youtube.com/watch?v=ViLReDIvk_A(accessed on 21 March 2024), https://www.youtube.com/watch?v=C0Rl4m38gOU(accessed on 21 March 2024). Additionally, 20 volunteers (10 males and 10 females, aged 24.1 ± 2.05 years) were separately recruited to rate the valence and arousal of the VR imagery materials before EEG data collection. Details and evaluation results of the video materials employed in the experiment are shown in Table 1. The content of the videos observed in the VR-2D and VR-3D groups was the same, with a consistent data format, a resolution of 4096 × 2048 dpi, a frame rate of 30 frames per second, and formatted in H.264 encoding. Videos were edited using Adobe Premiere software 2022. The difference between the VR-2D group and the VR-3D group was the removal of parallax between left eye images and right eye images in the VR-2D group. As shown in Figure 3, eight VR videos corresponding to the respective quadrants were selected.

### 2.5. Experiment Apparatus

All participants voluntarily took part in the experiment and received comprehensive information regarding the research objectives, experimental procedures, and associated risks. Prior to the experiment, participants signed a written informed consent form and received appropriate compensation after the experiment’s completion. The participants were also informed that they could stop the experiment at any time without any consequences. The experimenters explained the SAM emotional scale to the subjects and how to fill out the self-assessment form. Next, the experimenters fitted the EEG cap on the subjects and applied conductive paste to ensure that the electrode impedance was below 5 kΩ. After the EEG equipment was set up, all participants had to wear VR devices. During the experiment, participants were instructed not to randomly move their heads or bodies to avoid errors caused by limb movement and interference from the VR headset with the EEG cap. Before the formal playing of video clips, subjects were asked to sit quietly in a chair for five minutes, allowing them to quickly adapt to the experimental environment and calm down, thus avoiding experimental errors due to excitement. During the official video playback, the subjects’ EEG signals were synchronously recorded.

Figure 4a illustrates the experimental procedure for each trial, consisting of a baseline phase and nine trials. The first trial is a test trial intended to familiarize subjects with the experimental procedure. The specific process for each trial is shown in Figure 4b. After the experiment started, eight VR videos were presented in random order across the eight trials. Each trial included three steps. First, a fixed plus sign is displayed for five seconds. Second, the subjects randomly watched one of the VR videos. Following each viewing, participants were prompted to subjectively assess their present emotional state. To mitigate potential artifacts stemming from frequent donning of the VR head-mounted display (VR HMD) and potential disruption to EEG electrode positioning, the SAM emotional scale was presented on a virtual screen. This functionality enabled subjects to complete the SAM emotional scale using the VR controller without the need to remove the VR HMD. The scoring and rest process took approximately three minutes. During this time, subjects can also recuperate from the preceding emotional induction phase to avoid fatigue. Each VR video lasts approximately three minutes. The experiment took approximately one hour to complete. The total duration of the experiment was kept concise to prevent fatigue in the subjects.

### 2.6. Data Preprocessing

Clean and high-quality EEG signals were obtained through preprocessing the experimental data using EEGLAB 2021.1, an open-source MATLAB 2021b toolbox renowned for its robust EEG preprocessing capabilities, feature extraction, and emotion recognition algorithms [29]. As depicted in Figure 5, the data preprocessing included the following steps: first, importing the raw EEG data into MATLAB; second, locating the electrode points; third, applying FIR bandpass filtering in the range of 0.1–75 Hz to minimize the introduction of artifacts and filtering out 48–52 Hz electrical noise; fourth, downsampling the data to 200 Hz; fifth, segmenting the data based on marker labels, and dividing each subject’s collected EEG data into eight trials, with each trial comprising approximately three minutes of video playback (video data). Subsequent preprocessing for each trial involved bad lead interpolation and bad segment removal, using independent component analysis to eliminate artifacts such as blinking, eye drift, and electromyographic noise.

To mitigate the impact of video duration on EEG analysis results, the EEG data from all subjects were truncated to equal lengths, with each subject’s EEG data (for each emotional state) having a total duration of 120 s, amounting to 120,000 data points. Additionally, to ensure the experimental data’s reliability, resting-state EEG signals were separately collected for subjects in both modes. The resting-state durations for both modes were 60 s each, totaling 60,000 data points, to check for significant differences in baseline brain activity between the two groups before formally watching the videos. After data preprocessing, purified EEG signals were obtained for the subsequent feature extraction steps.

### 2.7. Data Analysis

#### 2.7.1. Questionnaire Analysis

For the data from the SAM scale, one-way analysis of variance was conducted using SPSS 27.0.1 statistical software (IBM Inc., Chicago, IL, USA) [30]. Independent two-sample *t*-tests were performed, and the EEG PSD was statistically analyzed with correction for multiple comparisons using the Benjamini-Hochberg False Discovery Rate (BHFDR). All results were reported at a significance level of 0.05.

#### 2.7.2. EEG Analysis

Utilizing Butterworth bandpass filters, each EEG segment was decomposed into four frequency bands (namely, θ[4~8Hz],α[8~13]Hz,β[13~30Hz] and γ[30~50Hz]). Subsequently, the PSD (measured in ${\mu V}^{2}/Hz$ units) [30] features of EEG signals in each frequency band were computed using fast Fourier transform (FFT). PSD $P(X_{i})$ can be defined according to Equation (1):(1)$$P\left(x_{i}\right)=\frac{1}{n}\sum_{i=1}^{n}{X(k)}_{i}^{2}$$
where $X_{k}$ represents the FFT of the EEG segment signal $x_{i(n)}$, k denotes the number of data segments (in this experiment, the number of data segments is 1), and n indicates the number of data points in each segment (in this experiment, the number of data points is 120,000).

We computed the PSD of each subject across different frequency bands, which reflects the distribution of signal power in the frequency domain [30,31]. Subsequently, we utilizde the PSD to compare differences in brain activation between the two modes.

As shown in Figure 6, the following brain regions and channels were selected for analysis in this study to explore the results of PSD differences in the θ,α,β and γ frequency bands across different brain areas: the frontal region (FP1, FP2, FZ, F3, F4, FCZ, FC3 and FC4), parietal region (C3, CZ, C4, CP3, CPZ, CP4, PZ, P3 and P4), temporal region (F7, F8, FT7, FT8, T7, T8, TP7, TP8, P7 and P8), and occipital region (OZ, O1 and O2).

To quantitatively assess the differences in EEG signals between the two modes, this study employed Cohen’s d as the measure of effect size. The formula for calculating Cohen’s d is as follows:(2)$$d=\frac{{\bar{x}}_{VR-3D}-{\bar{x}}_{VR-2D}}{\sqrt{\frac{\left(n_{1}-1\right){SD}_{1}^{2}+\left(n_{2}-1\right){SD}_{2}^{2}}{n_{1}+n_{2}-2}}}$$
where ${\bar{x}}_{VR-3D}$ and ${\bar{x}}_{VR-2D}$ represent the mean values of specific brain regions under the VR-2D and VR-3D modes, respectively, and $n_{1}$ and $n_{2}$ denote the sizes of the two sample groups, with ${SD}_{1}$ and ${SD}_{2}$ being their respective standard deviations.

Based on the magnitude of Cohen’s d value, the significance of the effect can be interpreted: a d value less than 0.2 indicates a small effect, greater than 0.5 suggests a medium effect, and greater than 0.8 denotes a large effect [32].

Furthermore, to quantitatively assess the extent of data support for the null hypothesis (H0) versus the alternative hypothesis (H1), this research incorporates the use of the Bayesian factor as a statistical instrument. Within the scope of this study, the null hypothesis stipulates that there are no significant differences in EEG signal activities across all brain regions and frequency bands between VR-2D and VR-3D environments. Conversely, the alternative hypothesis posits the existence of at least one brain region wherein EEG signal activity exhibits significant differences across at least one frequency band when comparing VR-2D to VR-3D environments.

By employing the Bayesian Information Criterion (BIC), as proposed by Jarosz et al. [33], an approximation of the Bayesian factor is calculated. This methodology facilitates a direct evaluation of the evidence supporting each hypothesis from the data, thus offering a quantified pathway for analysis.

The level of data support for the alternative hypothesis is stratified based on the inverse of the Bayesian factor: an inverse value ranging from 1 to 3 signals weak support or merely anecdotal evidence for the alternative hypothesis. An inverse falling between 3 and 10 indicates positive or substantial support for the alternative hypothesis. Meanwhile, an inverse value from 10 to 20 denotes strong support for the alternative hypothesis.

#### 2.7.3. Feature Selection

Given that small datasets in multidimensional feature spaces are prone to overfitting, feature selection was conducted to prevent this and enhance EEG-based emotion recognition performance. Recursive feature elimination (RFE) iteratively selects features by progressively considering smaller subsets of features. Initially, an estimator is trained on the full feature set, and the importance of each feature is assessed using attributes such as coef_ or feature_importances_. Subsequently, the least important features are eliminated from the current set. This process iterates recursively on the pruned set until the desired number of features is achieved. Specifically, to increase the number of training samples, the original EEG signals were divided into T= 1 s long, nonoverlapping segments (yielding approximately 14,400 samples per subject), with each segment assigned the same label as the original EEG signal. Next, each segment was decomposed into five frequency bands (namely, δ[1~4Hz],θ[4~8Hz],α[8~13]Hz,β[13~30Hz] and γ[30~50Hz]) using Butterworth bandpass filters. DE features were extracted from each frequency band with a 0.5 s window. For each segment, the DE feature rates of the five frequency bands were calculated, yielding 30 × 5 = 150 features. The RFECV function in sklearn [34] was employed to perform RFE within a cross-validation loop to determine the optimal number of features. In this study, the Support Vector Machine (SVM) [35] model was utilized for feature selection. The SVM model was fitted using 150 features, calculating the importance of each feature to determine the optimal number. Subsequently, only the selected DE features were used to classify brain activation states using an SVM. DE features have been suggested to be among the most effective for emotion recognition [36].

DE features are used to measure the complexity of EEG signals. The DE feature h(Z) is defined as:(3)$$h\left(Z\right)=-\int f\left(z\right)log\left(f\left(z\right)\right)dz$$
where z is a random variable and f(z) is the probability density function of z. If z follows a Gaussian distribution $N(\mu ,\delta^{2})$, then the DE feature in Equation (1) is calculated using the following formula:(4)$$h\left(Z\right)=\int_{-\infty }^{+\infty }\frac{1}{\sqrt{2\pi \delta^{2}}}exp\frac{{\left(z-\mu \right)}^{2}}{{2\delta }^{2}}log\frac{1}{\sqrt{2\pi \delta^{2}}}exp\frac{{\left(z-\mu \right)}^{2}}{2\delta^{2}}dz=\frac{1}{2}log2\pi e\delta$$
where e and δ represent Euler’s constant and the standard deviation of Z, respectively.

#### 2.7.4. Machine Learning-Based Classification of Brain Activation States

Because our collected dataset has its own characteristics, SVM is not biased towards VR-2D or VR-3D data. Additionally, SVM is widely used in classification tasks and has shown good performance across various domains. Therefore, we chose the SVM-based method as the initial classification approach. Subsequently, to further improve the accuracy of EEG emotion recognition, we introduced deep learning methods.

The ratings given by the subjects for each video based on arousal and valence (1–9) were used as labels, and these labels were divided into two binary classification problems based on thresholds of 5 on both the arousal and valence dimensions. A support vector machine (SVM) was employed to classify the DE features of EEG signals in both modes to verify whether EEG signals under these two modes could be differentiated. Evaluation metrics included average accuracy, precision, recall, specificity, and F1 score. To optimally utilize the data and select the best parameter combination, this study adopted a dual ten-fold cross-validation method. The outer layer utilized StratifiedKFold for data partitioning to fully exploit the data, while the inner layer employed a grid search algorithm (grid search, GS) to obtain the best SVM parameter combination. The search range for parameter C was set to [1×10^−5^, 11] and that for gamma was set to [1×10^−5^, 1], with both ranges divided into 10 equal parts. The search range for the kernel parameter was set to ‘linear’ and ‘rbf’.

#### 2.7.5. Deep Learning-Based Classification of Brain Activation States

To delve deeper into the differences in EEG signals for emotional arousal between the two modes and effectively integrate the spatial-frequency-temporal information of EEG, Figure 7 illustrates the proposed hybrid model that combines DE, CNN, and LSTM, namely, the DE-CRNN EEG emotion recognition architecture. This architecture comprises six parts: original EEG signals, original EEG trials, EEG segments, the 4D spatial-frequency-temporal structure, the CRNN, and EEG classification results. Below, the details of the 4D spatial-frequency-temporal structure, CRNN, and classifier are sequentially described.

This study evaluates the performance of the EEG emotion recognition method using the established protocol by Li et al. [37]. Specifically, ten-fold cross-validation was conducted for each subject, with their individual performance represented by the average classification accuracy (ACC) and standard deviation (STD). The mean ACC and STD across all subjects signify the final performance of this model.

Training the DE-CRNN model on an NVIDIA GTX 3090 GPU (NVIDIA Corporation, Santa Clara, CA, USA). Adam optimizer is used, with the learning rate and epochs set to 0.001 and 100, respectively.

##### 4D Spatial-Frequency-Temporal Structure

Assume that the original EEG segment is represented as a feature vector $F_{n}\in R^{mrT}$, where m and r denote the number of channels and the sampling rate of the original EEG signal, respectively. For each EEG segment, DE features for each frequency band are calculated using 0.5 s nonoverlapping windows. Thus, the 3D feature tensor $P_{n}\in R^{mf2T}$ is extracted from the original EEG segment; n=1,2,⋯,N, where N is the total number of samples, f is the number of frequency bands, and 2T represents twice the length of the segment. In this paper, f is set to 5. To effectively utilize the spatial structure information of electrodes, all m electrode channels are organized into a compact 2D map of size 8 × 6. Different frequency bands’ 2D maps are then stacked into a 3D array to effectively integrate their complementary information. Therefore, the 3D feature tensor is ultimately transformed into a 4D feature tensor representation $X_{n}\in R^{hwf2T}$, where h and w are the height and width, respectively, of the 2D map. In this study, h = 8 and w = 6 are set.

##### CRNN

For the 4D spatial-frequency-temporal representation $X_{n}$, spatial and frequency information is extracted from each of its temporal slices through a CNN module, as depicted in Figure 7. The module comprises four convolution blocks, a max-pooling layer, and a fully connected layer. The ReLU activation function is applied to all convolution layers. A dropout layer with a 25% dropout probability follows the pooling layer to prevent overfitting and enhance the model’s generalization capability. The output from the dropout layer is flattened and fed into a fully connected layer with 64 units. Therefore, for each temporal slice $S_{i}$, the final output $P_{i}\in R^{64}$ serves as its representation in spatial and frequency.

Due to the temporal variations between different temporal slices containing valuable temporal information for emotion recognition, LSTM is employed to learn the temporal information of each temporal slice in the CNN output, thereby achieving more accurate emotion classification, an LSTM module is utilized to learn the temporal information from the time slices of the CNN output results. The LSTM module takes the output sequence $P_{n}=(P_{1},P_{2},\cdots ,P_{2T})$ from the CNN, where $P_{i}\in R^{64}$ and i=1,2,⋯2T. In Figure 7, an LSTM layer with 64 memory units is utilized to explore the temporal dependencies within the segments. The output of the LSTM layer is calculated as follows:(5)$$i_{t}=\sigma \left(w_{i}\cdot \left[h_{t-1},x_{t}\right]+b_{i}\right)$$
(6)$$f_{t}=\sigma \left(w_{f}\cdot \left[h_{t-1},x_{t}\right]+b_{f}\right)$$
(7)$$\tilde{c_{t}}=tanh\left(w_{c}\cdot \left[h_{t-1},x_{t}\right]+b_{c}\right)$$
(8)$$c_{t}=f_{t}\cdot c_{t-1}+i_{t}\cdot_{c_{t}}^{\sim }$$
(9)$$o_{t}=\sigma \left(w_{o}\left[h_{t-1},x_{t}\right]+b_{o}\right)$$
(10)$$h_{t}=o_{t}\cdot tanh\left(c_{t}\right)$$
(11)$$y_{t}=w_{y}\cdot h_{t}+b_{y}$$
where σ represents the sigmoid activation function, and i,f,o,c and h represent the input gate, forget gate, output gate, update cell, and hidden state, respectively. $y_{t}$ denotes the output of the gated recurrent unit at time t. w and b are the weight matrices and bias terms, respectively.

The final output is obtained from the last LSTM node. Therefore, the ultimate representation of the EEG segment is $Y_{n}\in R^{64}$, which integrates the frequency, spatial, and temporal information of $X_{n}$.

##### Classifier

From the ultimate representation $Y_{n}$ of the EEG signal, a fully connected layer and a softmax activation function are utilized to predict the label of the 4D feature $X_{n}$, which is described as follows:(12)$$P_{c}=softmax\left(W_{c}Y_{n}+b_{c}\right)$$
where $W_{c}$ and $b_{c}$ are learnable parameters and $P_{c}\in R^{c}$ represents the probability of EEG segment $X_{n}$ belonging to emotion category C.

## 3. Experimental Results

### 3.1. Results of Subjective Data Evaluation

Based on the statistical analysis of the SAM subjective scale, significant group main effects for both valence (F = 4.300, *p* = 0.039) and arousal (F = 6.150, *p* = 0.013) were observed between the VR-2D group and the VR-3D group. The valence and arousal data are detailed in Table 2, demonstrating that the VR-3D group has a notably higher average emotional arousal than the VR-2D group. In summary, participants demonstrated proficiency in accurately discerning the emotional content of the experimental materials, indicating successful emotion induction during the study.

### 3.2. Comparison of Brain Region Topography Maps

Brain topography maps visually illustrate the spatial distribution of EEG signals, showcasing high (red) and low (blue) power levels and trends across various brain regions. This method effectively highlights differences in EEG signals. As depicted in Figure 8, brain topography maps for the four quadrants in the four frequency bands (θ,α,β and γ) were plotted using MATLAB 2021b software for both VR-2D mode and VR-3D mode. In bands θ,α,β and γ, the overall brain area energy of the VR-3D group appeared visually higher compared to that of the VR-2D group. Higher energy regions were predominantly concentrated in the frontal, temporal, and occipital regions.

Subsequently, the study conducted a comparative analysis of brain activity between groups. As illustrated in Figure 9, the PSD in various brain regions of the VR-3D group appeared noticeably higher than that in the VR-2D group.

The statistical analysis involved independent two-sample t-tests with correction for multiple comparisons using the BHFDR for the analysis of EEG PSD. Firstly, with Levene’s test yielding a *p*-value greater than 0.05, we accepted the assumption of homogeneity of variances. Subsequently, the comparison of resting-state EEG signals between the two mode groups indicated no significant differences in the resting EEG signals across four frequency bands in each brain region for both groups, with *p*-values greater than 0.05. Consequently, subsequent EEG signal analyses for both groups were conducted at the same level, ensuring the reliability of the objective data. As shown in Table 3 and Table 4, significant differences in EEG signals during video viewing were observed between the two groups in each brain region across the four frequency bands. Compared to the VR-2D group, the VR-3D group exhibited higher brain activation in the frontal, temporal, and occipital regions.

Table 5 and Table 6 present analyses of Cohen’s d values across different frequency bands for the frontal and parietal regions, as well as the temporal and occipital regions, among participants watching videos in VR-2D and VR-3D groups. The experimental findings reveal significant variability in EEG activity across various brain regions and frequency bands, with the VR-3D mode eliciting more pronounced brain electrical responses compared to the VR-2D mode.

Table 7 and Table 8 provide a detailed analysis of the Bayesian factor inverses for the frontal, parietal, temporal, and occipital regions in different frequency bands while participants from VR-2D and VR-3D groups watched videos. They offer compelling evidence in support of the alternative hypothesis. The experimental results indicate that compared to the VR-2D environment, the VR-3D environment elicits stronger brainwave responses.

Overall, the brainwave energy induced by VR-3D videos is significantly higher than that of VR-2D videos (mainly concentrated in the frontal, temporal, and occipital regions of the brain).

### 3.3. Results of Feature Selection

During the EEG-based emotion classification process, each subject has 14,400 samples, with each sample comprising 30 × 5 = 150 features. As illustrated in Figure 10, the optimal number of features is determined based on their effectiveness in predicting the model’s output. Ultimately, 91 features were selected for the arousal dimension in VR-2D mode, 100 features for the arousal dimension in VR-3D mode, 139 features for the valence dimension in VR-2D mode, and 78 features for the valence dimension in VR-3D mode for classifying brain activation states.

### 3.4. Results of Machine Learning Classification

The results for average accuracy, average precision, average recall, average specificity, and average F1 score based on the SVM classifier are shown in Table 9. Figure 11 and Figure 12 depict the ROC curves and confusion matrices of EEG emotion recognition experiment results for valence and arousal dimensions across both modes. The experimental results indicate that all metrics in the VR-3D mode outperform those in the VR-2D mode.

The receiver operating characteristic (ROC) curves and the area under the ROC curve (AUC) for each subject were calculated. The ROC curve directly reflects the strengths and weaknesses of the sensitivity and specificity metrics. The AUC, being the area under the ROC curve, provides a direct observation of the classifier’s performance. A larger AUC value indicates better performance. This paper applied the microaverage method to obtain the ROC curves, and Figure 11 displays the ROC curves and AUC values for the EEG emotion recognition experiment results across the dimensions of valence and arousal in the two modes.

Additionally, the percentages and corresponding sample numbers of the confusion matrix for emotion states based on the SVM algorithm are shown in Figure 12. In the confusion matrix, rows represent true values, and columns represent predicted values.

### 3.5. Results of Deep Learning Classification

Table 10 illustrates the average accuracy results obtained from the DE-CRNN model. Figure 13a,b display the performance of each subject across the dimensions of valence and arousal in the two modes. Figure 14 shows the ROC curves a for the EEG emotion recognition experiment results in these two modes for the dimensions of valence and arousal. The experimental outcomes demonstrate that all metrics in the VR-3D mode surpass those in the VR-2D mode.

Figure 14 exhibits the ROC curves and AUC values for the EEG emotion recognition experiment results for valence and arousal dimensions across the two modes. ‘sub1’ represents the ROC curve and AUC value for Subject 1’s results, and the pattern is similar for other subjects. Notably, each subject’s AUC value reaches a high level, indicating that the model is robust.

### 3.6. Interpretability of Machine Learning

To comprehend the workings of machine learning models, an increasing number of researchers are exploring interpretable methods for machine learning. In recent years, local interpretability methods have gained widespread attention, with many researchers proposing approaches for local explanations, effectively analyzing the model’s predictive principles on individual samples. Currently, the most mainstream interpretable methods based on local explanations are Shapley values and counterfactual explanations. Shapley values indicate the contribution of each individual feature to the model’s output. In this study, we adopt an interpretability method based on Shapley values.

As depicted in Figure 15, explanations for all sample predictions are presented in two graph types. The first type is the standard bar chart derived from the average absolute values of the shap values for each feature, showing the contribution of each feature to the prediction probability. The second type comprises scatter plots that simply depict the shap values for each feature of each sample. The color coding in these plots illustrates the relationship between the magnitude of feature values and their predictive impact and shows the distribution of these feature values. Specifically, (1) Figure 15a indicates that in the valence dimension under VR-2D mode, Features 3, 2, and 76 are important; (2) Figure 15b shows that in the valence dimension under VR-3D mode, Features 45, 73, and 2 are significant; (3) Figure 15c reveals that in the arousal dimension under VR-2D mode, Features 1, 63, and 10 are important; and (4) Figure 15d demonstrates that in the arousal dimension under VR-3D mode, Features 18, 4, and 12 are important.

In each subfigure of Figure 15, the right half demonstrates that the redder the color is, the higher the value, and vice versa. Additionally, when the value is positive, it contributes to the probability of a class 0 prediction.

## 4. Discussion

This paper conducted statistical analysis on the SAM scale and the EEG PSD for four emotions and employed machine learning and deep learning techniques to classify EEG-based emotional responses induced under VR-2D and VR-3D modes. The aim was to explore the impact of these two modes on brain activity during the arousal process of different emotions. The following sections provide a more detailed discussion.

### 4.1. Brain Activity Differences

The study findings demonstrate that in both the VR-2D mode and VR-3D mode, the VR-3D mode exhibits more significant brain activation states. The regions with the most substantial differences in brain activation across the four emotions are located primarily in the frontal, temporal, and occipital regions, consistent with findings from previous studies. Perry et al. [38] confirmed that the frontal region is closely related to emotions. The hippocampus, which is located in the deep grooves of the temporal region, plays a significant role in emotional processing and cognitive functions. The occipital region, which is responsible for visual information processing in the brain, is associated with increased αactivity during multimedia presentations (videos and images) [23].

### 4.2. Brain Activation State Classification

In this study, first, the original EEG signals were segmented into nonoverlapping segments of T = 1 second to increase the number of training samples. Second, recursive feature elimination was employed for feature selection, followed by emotion classification using an SVM with dual ten-fold cross-validation. The results demonstrated effective classification in both modes, with notably higher EEG-based emotion recognition accuracy in the VR-3D mode. Last, to delve deeper into the differences in EEG signals for emotional arousal between the two modes and to effectively integrate the EEG’s spatial-frequency-temporal information, this paper proposed the DE-CRNN model for EEG-based emotion classification. The results indicated more significant brain activation in the VR-3D mode, and the high AUC values for each subject in the ROC curves underscored the robustness of the DE-CRNN model.

### 4.3. Future Work

Considering some limitations of this study, first, EEG signals are susceptible to interference. To mitigate artifacts such as ocular and myoelectric noise and uphold the integrity of the experimental data, subjects were instructed to refrain from rotating their heads during the tasks. Although this restriction limited participants from viewing the full 360° VR videos, thereby somewhat attenuating the immersive experience, it was necessary to ensure data validity. Future study endeavors should contemplate validating the outcomes presented herein through magnetic resonance imaging (MRI) or dedicated MRI scans, thereby delving deeper into the distinctions in brain activation across the two VR modalities. Second, the current study is limited to binary classification in the dimensions of valence and arousal. Future study aims to delve into classifying emotions across four distinct categories. Third, this paper focused on EEG-based emotion classification across all channels and frequency bands. Future studies will separately investigate specific channels and each frequency band to identify key channels and frequency bands for EEG-based emotion recognition. Fourth, the participants in this study consisted of undergraduate and graduate students. In the future, we plan to recruit participants from different age groups to further enhance the generalizability of the experiment. Fifth, although the deep learning model proposed in this study can effectively capture the temporal information of EEG signals, Graph Neural Networks (GNNs) are better suited for handling spatial information. Therefore, in the future, we will consider using GNNs to extract temporal information from EEG signals.

## 5. Conclusions

This study aimed to explore the disparities in brain activation between VR-2D and VR-3D modes by presenting emotional videos across four emotion categories to elicit EEG-based emotional responses in participants. For this purpose, we conducted statistical analyses on the SAM scale and the EEG PSD for the four emotion categories, elucidating the impact of these two sensory modes on brain activity during different emotional arousal processes. We also classified EEG-based emotions induced in both modes using machine learning and deep learning techniques, achieving favorable classification results, which further validated the neurophysiological differences between the two modes. Our study demonstrated more significant brain activation in the VR-3D mode. The contribution of features to the prediction probability was discussed using a Shapley value-based machine learning interpretability approach. The findings of this study offer novel insights into the neural mechanisms underlying continuous emotion induction in VR-2D and VR-3D modes. This contributes significantly to the ongoing assessment of the interplay and ramifications of emotion induction across both modes, thereby facilitating advancements in affective brain–computer interface technologies.

## Figures and Tables

**Figure 1 brainsci-14-00326-f001:**
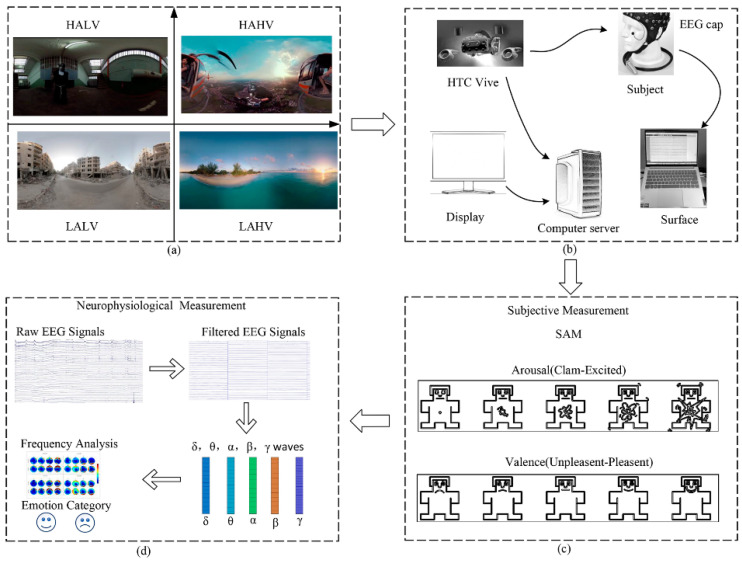
Emotional Induction and Analysis Process in VR: (**a**) Anticipated Emotional States and Emotional Induction Materials, (**b**) Experimental Setup, (**c**) Subjective Data Processing and Analysis, and (**d**) Neurophysiological Data Processing and Analysis.

**Figure 2 brainsci-14-00326-f002:**
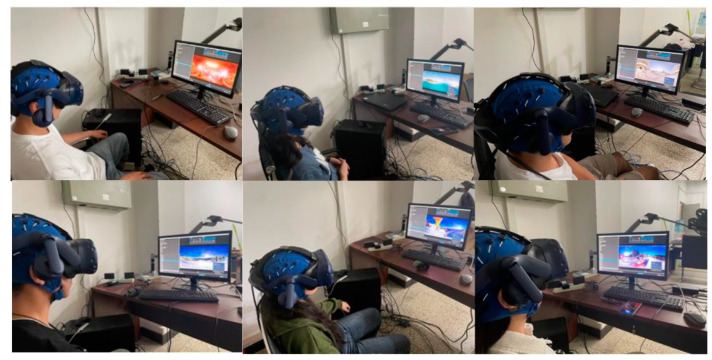
Schematic of experimental equipment connections and a sample screenshot of a participant.

**Figure 3 brainsci-14-00326-f003:**
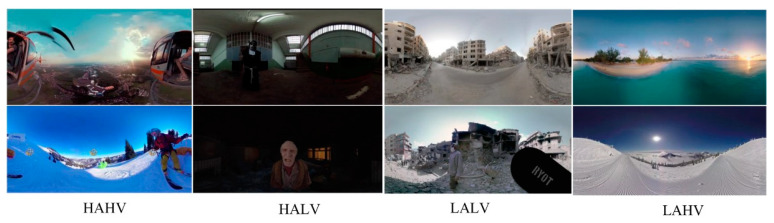
VR videos used in the experiment. The four columns correspond to the emotions HAHV, HALV, LALV, and LAHV. Each column, from top to bottom, includes ‘Tomorrowland 2014’, ‘Speed Flying’, ‘The Real Run’, ‘The Conjuring 2’, ‘The Earthquake Site’, ‘The Nepal Earthquake Aftermath’, ‘Instant’.

**Figure 4 brainsci-14-00326-f004:**
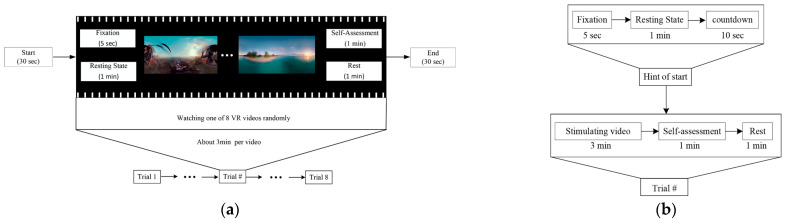
(**a**) Experimental procedure. The experiment comprises two parts: the emotional task and the subjective rating. Both the VR-2D group and VR-3D group consisted of nine trials each, where the first was an experimental trial and the remaining eight were formal trials; (**b**) experimental Procedure for Each Trial.

**Figure 5 brainsci-14-00326-f005:**
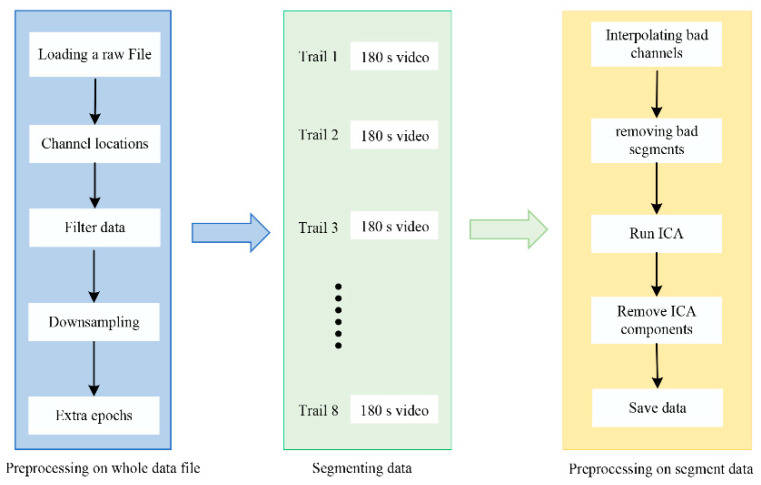
Flowchart of the preprocessing procedure for a subject’s original EEG signal.

**Figure 6 brainsci-14-00326-f006:**
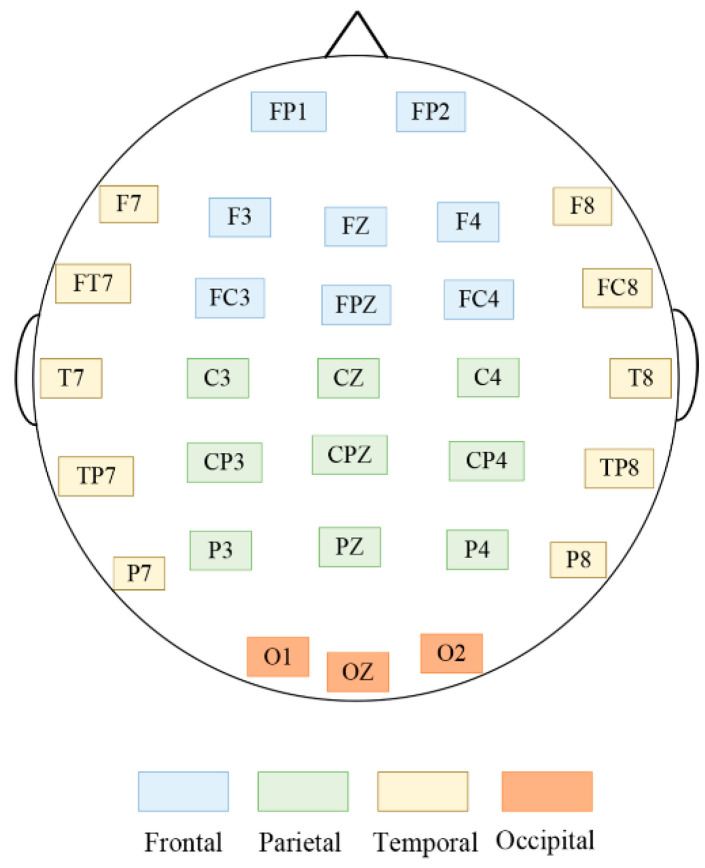
EEG electrode distribution diagram.

**Figure 7 brainsci-14-00326-f007:**
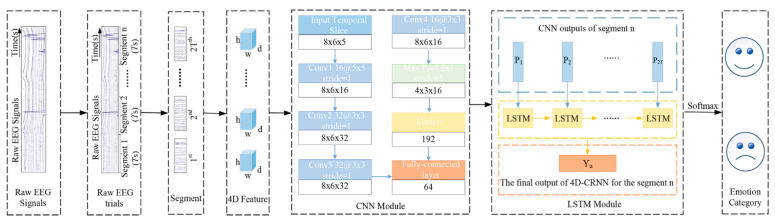
Overall architecture of EEG emotion recognition based on DE-CRNN.

**Figure 8 brainsci-14-00326-f008:**
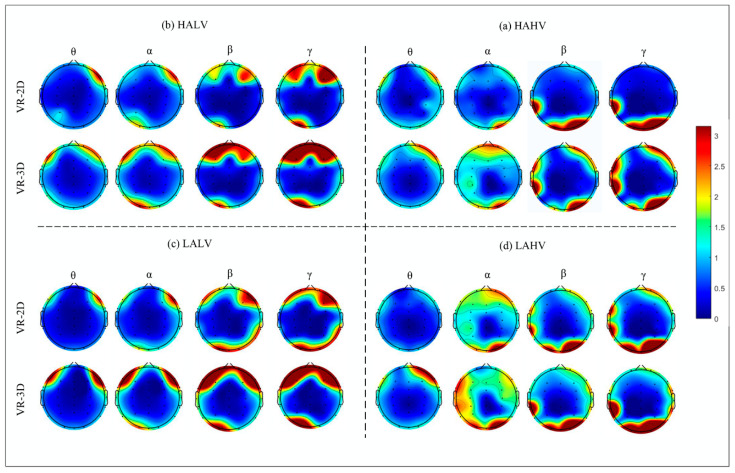
Brain Topography Maps of Different Emotions in θ, α, β, and γ Frequency Bands for Power Density in VR-2D and VR-3D Groups.

**Figure 9 brainsci-14-00326-f009:**
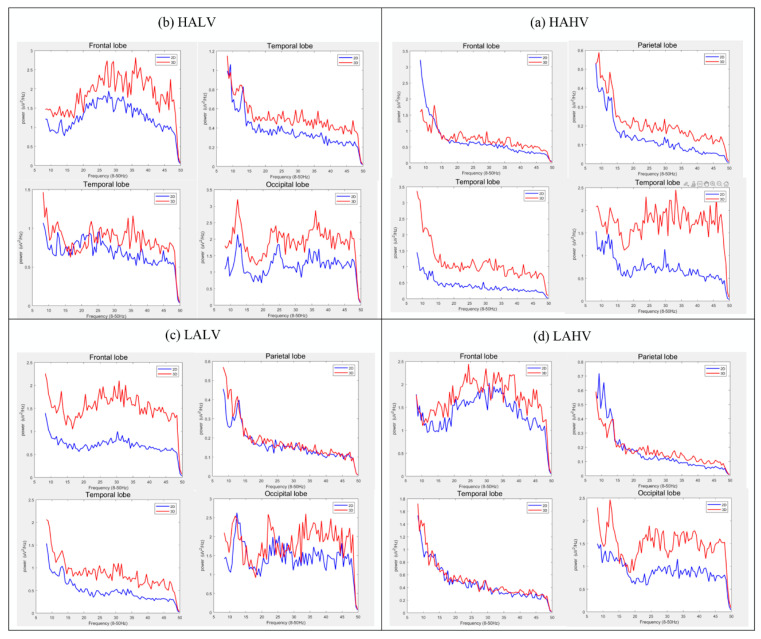
Comparison of Differences in PSD in θ, α, β, and γ Frequency Bands across Frontal, Temporal, Parietal, and Occipital Regions for Different Emotions in the VR-2D and VR-3D Groups.

**Figure 10 brainsci-14-00326-f010:**
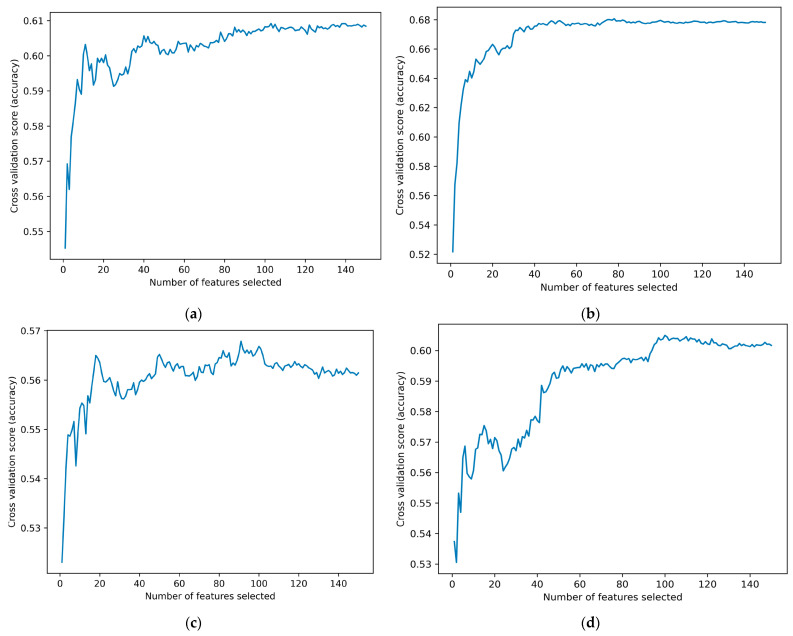
Feature Selection Graph. (**a**) The feature selection graph for the valence dimension under the VR-2D mode; (**b**) the feature selection graph for the valence dimension under the VR-3D mode; (**c**) the feature selection graph for the arousal dimension under the VR-2D mode; (**d**) the feature selection graph for the arousal dimension under the VR-3D mode.

**Figure 11 brainsci-14-00326-f011:**
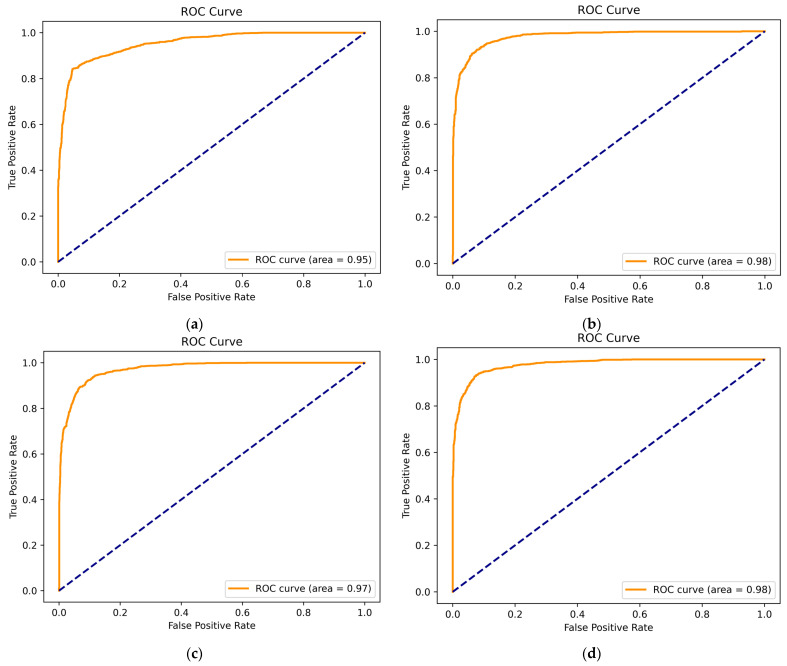
Microaverage ROC curve for subjects. (**a**) The ROC curve for the valence dimension under the VR-2D mode; (**b**) the ROC curve for the valence dimension under the VR-3D mode; (**c**) the ROC curve for the arousal dimension under the VR-2D mode; (**d**) the ROC curve for the arousal dimension under the VR-3D mode.

**Figure 12 brainsci-14-00326-f012:**
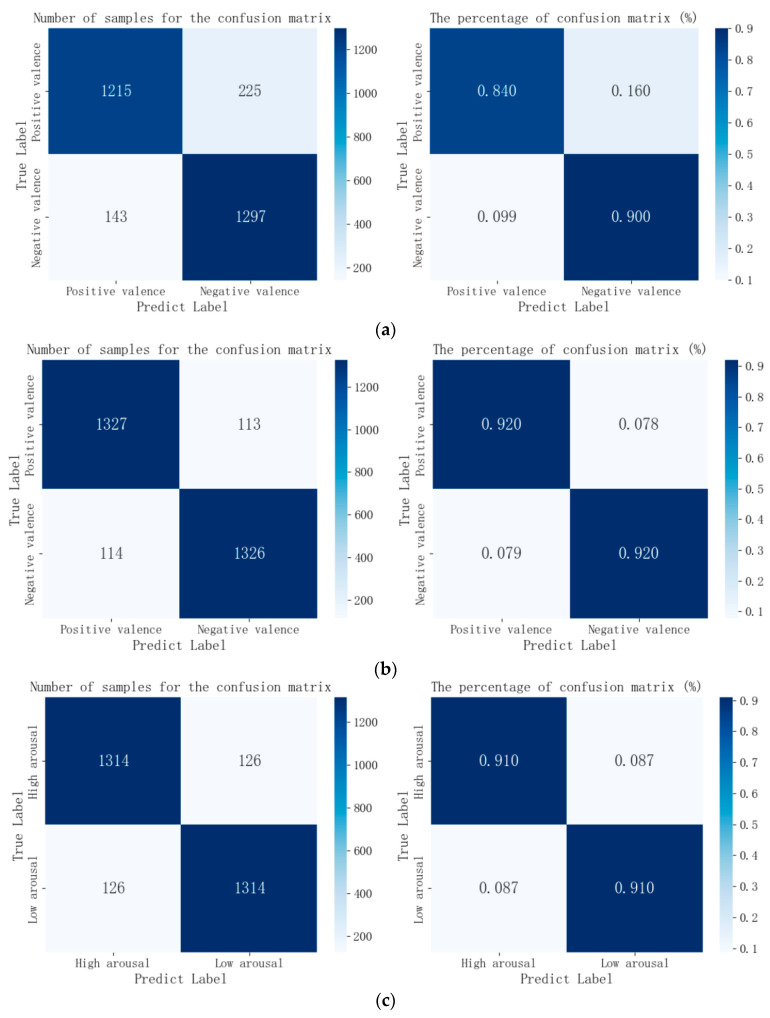
Confusion matrix for subjects. (**a**) The confusion matrix for the valence dimension under the VR-2D mode; (**b**) the confusion matrix for the valence dimension under the VR-3D mode; (**c**) the confusion matrix for the arousal dimension under the VR-2D mode; (**d**) the confusion matrix for the arousal dimension under the VR-3D mode.

**Figure 13 brainsci-14-00326-f013:**
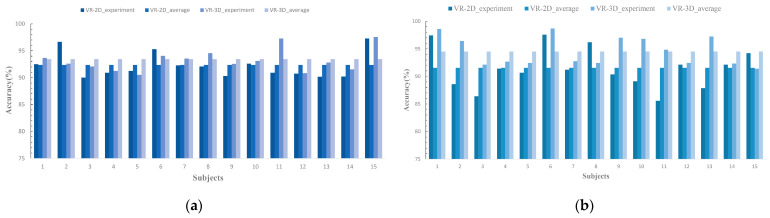
Performance of the DE-CRNN on the dataset. (**a**) Valence dimension; (**b**) Arousal dimension.

**Figure 14 brainsci-14-00326-f014:**
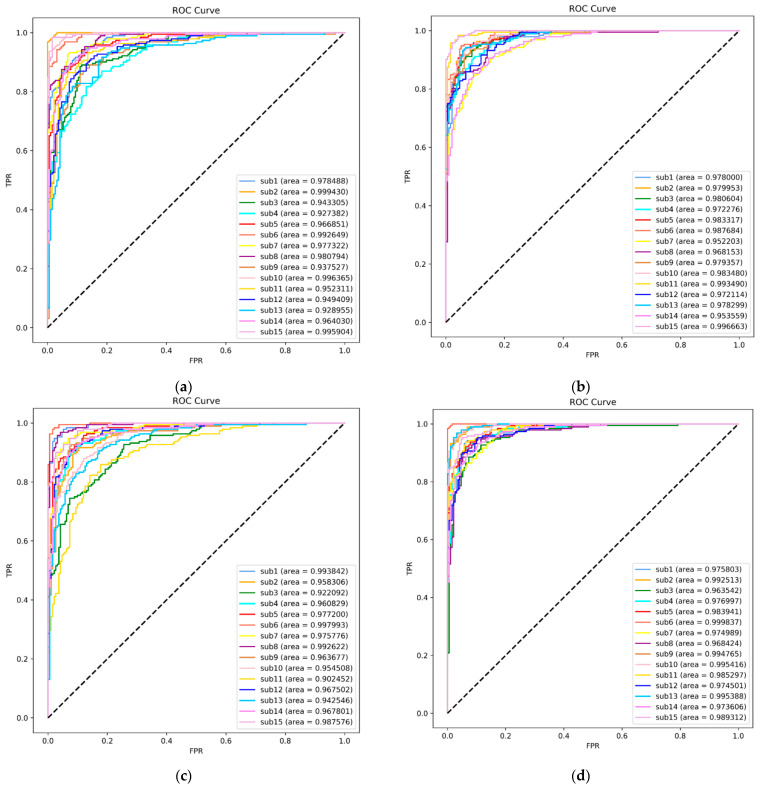
Microaverage ROC curve for subjects. (**a**) The ROC curve for the valence dimension under the VR-2D mode; (**b**) the ROC curve for the valence dimension under the VR-3D mode; (**c**) the ROC curve for the arousal dimension under the VR-2D mode; (**d**) the ROC curve for the arousal dimension under the VR-3D mode.

**Figure 15 brainsci-14-00326-f015:**
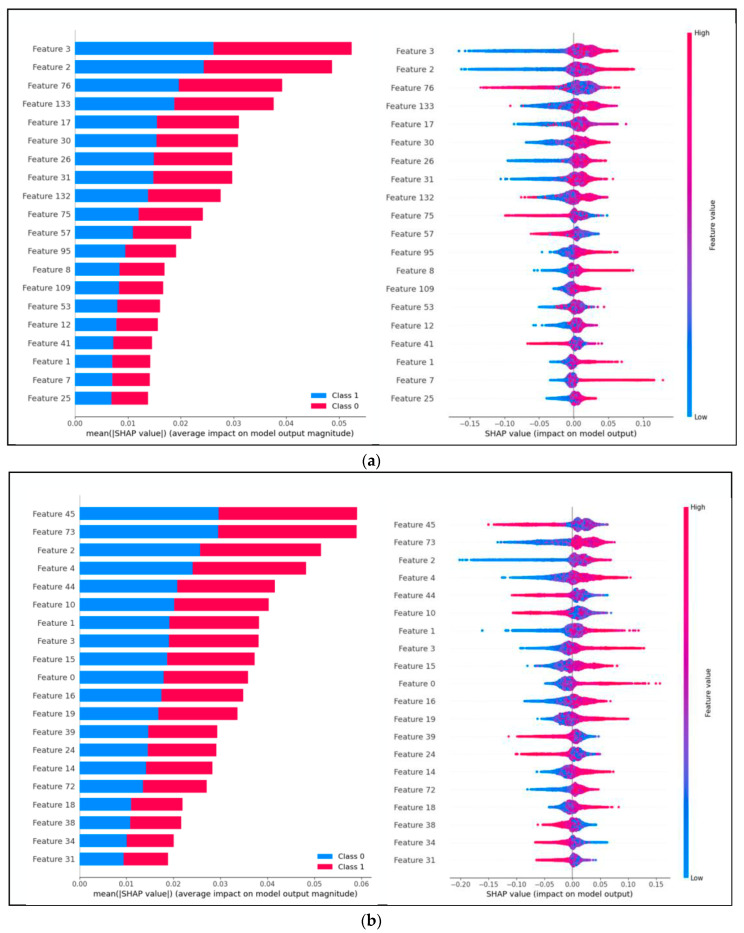
Model interpretability. (**a**) Explanation of sample predictions in the valence dimension under VR-2D mode; (**b**) explanation of sample predictions in the valence dimension under VR-3D mode; (**c**) explanation of sample predictions in the arousal dimension under VR-2D mode; (**d**) explanation of sample predictions in the arousal dimension under VR-3D mode.

**Table 1 brainsci-14-00326-t001:** Valence and arousal assessment results of video materials.

	Name of Video	Valence	Arousal
HAHV	Tomorrowland 2014	6.56 ± 0.98	6.36 ± 1.03
	Speed Flying	6.94 ± 0.56	7.16 ± 0.35
HALV	The Real Run	3.14 ± 0.89	7.89 ± 0.79
	The Conjuring 2	3.26 ± 0.64	7.63 ± 1.16
LALV	War Zone	2.49 ± 0.81	3.95 ± 0.65
	The Nepal Earthquake Aftermath	2.67 ± 0.62	3.34 ± 0.78
LAHV	Instant Caribbean Vacation	6.84 ± 0.58	3.43 ± 0.78
	The Mountain Stillness	6.16 ± 0.83	2.13 ± 0.63

**Table 2 brainsci-14-00326-t002:** Statistical Scores of the SAM Subjective Scale.

		VR-2D		VR-3D	
		Valence	Arousal	Valence	Arousal
HAHV	Tomorrowland 2014	5.73 ± 0.97	5.36 ± 1.19	6.66 ± 0.93	6.76 ± 1.06
	Speed Flying	6.12 ± 0.76	6.83 ± 0.49	6.34 ± 0.58	7.36 ± 0.43
HALV	The Real Run	2.99 ± 0.83	6.72 ± 0.86	3.24 ± 0.78	7.26 ± 0.63
	The Conjuring 2	3.15 ± 0.62	7.29 ± 1.23	3.46 ± 0.79	7.78 ± 1.19
LALV	War Zone	2.26 ± 0.87	3.14 ± 0.57	2.37 ± 0.76	3.72 ± 0.62
	The Nepal Earthquake Aftermath	2.18 ± 0.60	3.04 ± 0.76	2.35 ± 0.83	3.45 ± 0.95
LAHV	Instant Caribbean Vacation	6.56 ± 0.61	3.12 ± 0.89	6.84 ± 0.64	3.34 ± 0.56
	The Mountain Stillness	6.03 ± 0.73	2.13 ± 0.61	6.78 ± 0.89	2.56 ± 0.85

**Table 3 brainsci-14-00326-t003:** Differences in frontal and parietal EEG signals of participants in VR-2D and VR-3D groups while watching videos.

	VR-2D vs. VR-3D							
	Frontal				Parietal			
	θ	α	β	γ	θ	α	β	γ
HAHV	*p* < 0.01	*p* < 0.01	*p* = 0.021	*p* = 0.042	*p* = 0.519	*p* = 0.738	*p* = 0.102	*p* = 0.677
HALV	*p* < 0.01	*p* < 0.01	*p* < 0.01	*p* < 0.01	*p* = 0.762	*p* = 0.196	*p* = 0.172	*p* = 0.123
LALV	*p* = 0.045	*p* = 0.703	*p* = 0.025	*p* = 0.034	*p* = 0.993	*p* = 0.438	*p* = 0.588	*p* = 0.778
LAHV	*p* < 0.01	*p* < 0.01	*p* = 0.312	*p* = 0.239	*p* < 0.01	*p* < 0.01	*p* = 0.736	*p* = 0.326

**Table 4 brainsci-14-00326-t004:** Differences in temporal and occipital EEG signals of participants in VR-2D and VR-3 D groups while watching videos.

	VR-2D vs. VR-3D							
	Temporal				Occipital			
	θ	α	β	γ	θ	α	β	γ
HAHV	*p* = 0.158	*p* = 0.035	*p* < 0.01	*p* < 0.01	*p* = 0.023	*p* = 0.021	*p* < 0.01	*p* < 0.01
HALV	*p* = 0.762	*p* = 0.112	*p* < 0.01	*p* < 0.01	*p* = 0.021	*p* = 0.016	*p* < 0.01	*p* < 0.01
LALV	*p* = 0.003	*p* = 0.485	*p* < 0.01	*p* < 0.01	*p* = 0.908	*p* = 0.292	*p* < 0.01	*p* < 0.01
LAHV	*p* < 0.01	*p* < 0.01	*p* = 0.001	*p* = 0.004	*p* < 0.01	*p* < 0.01	*p* < 0.01	*p* < 0.01

**Table 5 brainsci-14-00326-t005:** Analysis of Cohen’s d values across frequency bands for frontal and parietal regions.

	VR-2D vs. VR-3D							
	Frontal				Parietal			
	θ	α	β	γ	θ	α	β	γ
HAHV	0.7471	0.7479	0.4766	0.4174	0.1409	0.0702	0.3261	0.0934
HALV	0.7051	1.6831	0.8680	0.8024	0.0645	0.4728	0.2799	0.3154
LALV	0.4289	0.0761	0.4761	0.2963	0.0015	0.2053	0.1504	0.0848
LAHV	1.8220	1.5044	0.2976	0.2400	0.8437	0.9114	0.0615	0.2040

**Table 6 brainsci-14-00326-t006:** Analysis of Cohen’s d values across frequency bands for temporal and occipital RRegions.

	VR-2D vs. VR-3D							
	Temporal				Occipital			
	θ	α	β	γ	θ	α	β	γ
HAHV	0.2808	0.3313	0.8222	0.6702	0.3601	0.3574	1.1944	0.9591
HALV	0.0787	0.2946	0.7062	0.6791	0.3574	0.2813	1.0174	1.0159
LALV	0.5998	0.1841	0.8994	0.8174	0.0315	0.2575	0.7087	0.9210
LAHV	1.0828	1.0003	0.6610	0.6098	0.9817	0.9359	1.2392	1.6038

**Table 7 brainsci-14-00326-t007:** Analysis of the inverse Bayes factor across frequency bands for frontal and parietal Regions.

	VR-2D vs. VR-3D							
	Frontal				Parietal			
	θ	α	β	γ	θ	α	β	γ
HAHV	8.2418	9.8136	2.1378	3.3922	1.9327	1.6788	2.9162	1.5967
HALV	6.9499	4.0660	7.6346	7.1981	1.6980	2.2863	2.6034	2.9135
LALV	3.0046	1.7328	3.4519	2.1691	1.3105	3.1097	1.9508	1.7704
LAHV	4.0279	5.4490	2.4841	2.6069	5.4494	6.3686	1.6997	2.6825

**Table 8 brainsci-14-00326-t008:** Analysis of the inverse Bayes factor across frequency bands for temporal and occipital regions.

	VR-2D vs. VR-3D							
	Temporal				Occipital			
	θ	α	β	γ	θ	α	β	γ
HAHV	2.7516	3.0619	9.2643	8.0725	2.3571	2.4163	10.9494	6.0785
HALV	1.7115	2.8157	4.5103	7.0920	1.2327	2.6327	5.6044	3.9815
LALV	4.2679	1.6134	9.2033	7.6985	1.4394	2.5769	6.4850	5.2657
LAHV	4.6451	5.6115	6.3832	3.2609	6.0414	6.2966	8.4348	10.2791

**Table 9 brainsci-14-00326-t009:** Performance metrics in VR-2D and VR-3D modes.

	VR-2D		VR-3D	
	Arousal	Valence	Arousal	Valence
Acc_mean ± STD (%)	90.09 ± 0.010	89.73 ± 0.007	91.56 ± 0.014	91.69 ± 0.003
Precision_mean ± STD (%)	89.85 ± 0.210	88.57 ± 0.013	90.63 ± 0.031	91.35 ± 0.007
Recall_mean ± STD (%)	90.53 ± 0.007	90.89 ± 0.019	91.31 ± 0.033	92.13 ± 0.002
Specificity_mean ± STD (%)	89.66 ± 0.024	88.89 ± 0.021	90.69 ± 0.037	91.24 ± 0.008
F1-score_mean ± STD (%)	90.17 ± 0.007	89.68 ± 0.006	90.97 ± 0.013	91.73 ± 0.002

**Table 10 brainsci-14-00326-t010:** Average accuracy and standard deviation in VR-2D and VR-3D modes.

ACC_Mean ± STD (%)		
	VR-2D	VR-3D
Arousal Dimension	91.56 ± 0.70	94.53 ± 0.67
Valence Dimension	92.34 ± 0.07	93.45 ± 0.63

## Data Availability

The data presented in this study are available on request from the corresponding author due to privacy.

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
