# Peer review of "Differential Brain Activation for Four Emotions in VR-2D and VR-3D Modes"

_brainsci, 2024, doi:10.3390/brainsci14040326_

Round 1

Reviewer 1 Report

Comments and Suggestions for Authors

1.       Use of the undisclosed abbreviations (HAHV/HALV/LALV/LAHV) and DE-CRNN in the abstract is not advised.

2.       “Specifically, the main research contributions of this paper are summarized as follows:”. You are advised to present innovative contributions of the paper.

3.       In the 'Discussion' section, the experiments of this paper are discussed.” I suppose that the results of the experiments rather the experiments were discussed in the Discussion section.

4.       Recent studies have widely explored”. You are advised to enumerate these studies.

5.       Research indicates”. Which research?

6.       Bilgin et al. designed low- and high-124 arousal emotional environments in both 2D and VR settings[25].”. The reference number is usually placed next to author names.

7.       “However, most of these studies have focused on traditional 2D and VR environments, with less extensive research on the differences between the VR-2D mode and the VR-3D mode.”

a.       Related work on VR-3D mode was not reviewed.

b.       Review of related work is short and not critical.

8.       The detailed steps are illustrated in Figure 1”. A presentation in Figure 1 cannot be named as the presentation of detailed steps of the framework. A graphical abstract of the paper was just presented in Figure 1. However, this is not a place to present the graphical abstract.

9.       Did the study receive a written agreement from the participants? This information was expected in subsection 3.1, but it was provided in the subsection 3.5.

10.   This paper processed the preprocessed EEG signals”. Can the paper process EEG signals??

11.   “Second, the EEG signals were preprocessed to obtain clean and high-quality signals. Third, power spectral density (PSD) and differential entropy (DE) features were extracted from the preprocessed EEG signals. Fourth, the data underwent statistical analysis.”. The steps defined in beginning of the section of Methods do not agree in presentation sequence in the section Data analysis, since the first step in Data analysis section is statistical analysis.

12.   The manuscript is full of inconsistencies. The careful proofreading and elaboration of the presented material is required.

Comments on the Quality of English Language

Moderate editing of English language is required

Author Response

1.Thank you very much for your feedback. We have made the necessary modifications to the abstract, ensuring clarity and comprehensibility of the terms used. We appreciate your help and support.

2.Thank you for your suggestion. We have summarized the main research contributions of this paper as innovative contributions. We appreciate your attention and guidance.

3.Thank you for your insightful review and constructive feedback. We have carefully considered your comments and have made the necessary revisions to address the concerns raised. 

4.Thank you for your feedback. We have included relevant studies to support our findings.” For example, Bilgin et al. designed low- and high-arousal emotional environments in both 2D and VR settings [24]. By recording and comparing EEG signals, the authors discovered that brainwave energy in the δ, θ, α, and β frequency bands was significantly higher in VR environments than in flat environments, leading to the conclusion that VR more effectively stimulates and modulates subjects' emotions compared to traditional flat displays. Ding et al. [25] investigated the emotional impact of watching four emotional movie clips in VR and 2D conditions, integrating subjective emotional experiences with objective physiological responses. The authors found that VR elicited stronger emotional experiences and physiological responses than the 2D environment. Yu et al. investigated the potential neural mechanisms of subjective viewing of positive and negative emotional videos in 2D and VR visual experiences using electroencephalography and found that the β and γ networks in the VR group exhibited greater global efficiency [26].”

5.Thank you for your insightful review and constructive feedback. We have carefully considered your comments and have made the necessary revisions to address the concerns raised. The section in question has been removed as per your suggestion

6.Thank you very much for reviewing our manuscript and providing valuable feedback. We have made revisions to the relevant sections based on your suggestions to improve the quality and clarity of the paper.

7.Thank you for your feedback. We have revised the manuscript to include relevant studies on the VR-3D mode and made adjustments to the review of related work accordingly. We appreciate your attention and guidance.

8.Thank you for your suggestion. We have revised the methods section and the caption of Figure 1 in the manuscript accordingly.

9. Thank you for your feedback. We did receive informed consent from the participants. We believe that placing this information in the summary of the experimental procedure better aligns with the logical sequence of the experiment. Your suggestion is greatly appreciated.

10. This paper has preprocessed the EEG signals

11. We greatly appreciate your careful attention. We have thoroughly reviewed the manuscript, and we have made revisions to the Methods section as you suggested.

12. Thank you for your feedback. We have carefully proofread the manuscript and ensured consistency throughout the presented material. Your input is greatly appreciated.

Reviewer 2 Report

Comments and Suggestions for Authors

36% duplicate content was present.

Manuscript written by the author was fine and good.

There is a flow in paper.

Related works need to be added for this study.

Did you identify any  different features for power spectral density (PSD) and differential entropy (DE) ?

Any specific reason for selecting subjects between 21 to 26 years?

Need more explanation about experimental setup. this induce the other researchers to cite the article.

Any specific reason for Butterworth bandpass filters to decompose each segment ?

Figure 10. Feature Selection Graph was not clear .

Figure 12. captions were very small. it was very difficult to read.

Figure 15.captions were very small. it was very difficult to read.

Author Response

1.Thank you for your feedback. We have carefully reviewed the manuscript and checked the originality of the text using Turnitin, with a similarity rate of 16%.

2.Thank you for your feedback. We have carefully reviewed the manuscript to ensure the coherence and logical flow of the content.

3.Thank you for your suggestion. We have added relevant literature to the manuscript.

4.Thank you for your inquiry. Yes, we identified distinct features for both power spectral density (PSD) and differential entropy (DE) in our study. We found that PSD captures spectral power distribution across different frequency bands, while DE characterizes the irregularity or complexity of EEG signals. These complementary features provided comprehensive insights into the brain dynamics under different experimental conditions.

5.Thank you for your inquiry. We selected participants aged between 21 and 26 because individuals within the same age range are likely to have more psychological and physiological similarities, thereby reducing the interference of individual differences on the experimental results. However, this choice also comes with limitations. In future work, we will consider collecting EEG data from participants of different age groups to further investigate the impact of age on the experimental outcomes. We appreciate your suggestion.

6.Thank you for your inquiry. Butterworth bandpass filters were chosen to decompose each segment due to their desirable frequency response characteristics, particularly their ability to provide a flat frequency response in the passband with minimal ripple. This ensures that the filtered signals retain important frequency components relevant to the analysis while effectively removing noise and artifacts outside the specified frequency range. Butterworth filters are also known for their stability and ease of implementation, making them a popular choice in EEG signal processing. We believe that the use of Butterworth bandpass filters enhances the quality of signal decomposition and contributes to the reliability of our analysis.

7.Thank you for your feedback. We have revised the figures to improve their clarity.

Reviewer 3 Report

Comments and Suggestions for Authors

Brief summary: This article supports that brain activation differences for four emotion categories within valence arousal remains to be explored. They compared differences in brain activation under EEG-based VR displays. Power spectral density computations highlighted brain activation patterns for different emotional states across VR-2D and VR-3D modes. These main findings could contribute to prediction oand understanding of emotions.

Please see my comments below.

Abstract:

- The abstract needs to be reworked as it is very difficult for novel readership to understand the aim of the study (which should be explicit: the aim of the study is to...), the methodology: clearly indicate the number of participants, who are these participants and the methods used. The main findings should be presented and the importance of these findings.

- At the present stages, there are many abbreviations in the abstract that have not yet been introduced.

Introduction:

- The concept of emotions is briefly introduced by lines 30 to 34. Major theories that brought forward the understanding of emotions in the context of neurosciences such as the work of Paul Ekman and Caroll Izard should be discussed to further understand what is meant by emotions. Ekman's model is stated much later in the discussion. It is suggested discuss it prior to entering in the notions of EEG and artificial intelligence.

- The citation for line 3 is incorrect as it is from an article that cites another paper on the subject: Emotion recognition from multi-channel EEG via deep forest IEEE J. Biomed. Health Inf., 25 (2) (2021), pp. 453-464

- VR-2D and VR-3D (line 83) are not discussed.

- Examples of uses of EEG and VR are abundant in the literature and are not discussed. Recent advances should be added to the introduction.

- In this current version of this introduction, it is unclear what the problematic is : how is this work important and what will it solves. This should be made more explicit for the readership.

- While the aim of the study is stated, hypothesis are not mentioned in the last paragraph.

The abbreviation VR is used across the introduction but is introduced in the related work section. The related work section should be part of the introduction (to follow MDPI's guidelines for original research).

Methods:

- Participants: Who are the volunteers, why undergraduates and graduate? Why 32 participants? How significants will he results be with this smaple? There is no mention of exclusion and inclusion criteria. It is very important to describe the sample and the rationale behind this sample. There is a wide range of literature describing divergences in emotional patterns depending on different physicial and psychological impairments. 

- How were the participants recruited?

- Measurements: The SAM validity and fidelity psychometrics propriety should be discussed.

- Virtual reality is redefined line 189.

- Feature selection: It is unclear why SVM was selected as the main model for the analysis.

- SK-Learn (scikitlearn) is unreferenced.

- StratifiedKFold partitioning has been used. The rational is to obtain the ''best svm parameter combination''. This should be further explored. Rational behind the model choice and limitations as well as the cross-validation strategy should be discussed.

- Choice of hyperparameter for the kernel should be based solely on ''the algorithm identified this parameter''. Choising RBF versus linear means that the data follows a radial distribution rather than a linear distribution. Why is that?

- Deep learning model: Again the choice of models, rational behind combining and cross validation should be reviewed.

- Fivefold cross-validation has been used, yet the standard in the literature is 10-fold validation. Why was 5 prefered over 10? This is a discussion point that needs to be adressed.

- Overall, the methodology lacks a clear explanation on the choice of technology, the rational behind these choice, the limitations of such models. This section is very technical and part of the mathematical explanations could be exported to supplementary material to ease the readership. 

Results:

- A vast array of results is presented and most of the figures (example: figure 15) are unreadable. This section should be reviewed.

- When presenting accuracies, are these the cross-validated accuracy. If so, does the numbers (example table 7) reflect averages of the cross validation or a sample value? Are the results presented for accuracy significant? If so, how is significancy established in your modelizations?

Discussion:

- Limitations of the AI models are not presented or discussed. Limitations are blended with future work which makes it difficult to appreciate the novelty and findings of this study. 

Conclusion:

- The conclusion does not currently highlight why this experiment is important and relevant to the field of study. This should be further explicited.

I encourage the authors to adress the above points to improve the manuscript. While the subject is interesting, it should be clearer to the readership : why your study is important, why is the methodology selected relevant and what are the limitations of such methodology.

Comments on the Quality of English Language

A lot of abbreviations are not introduced, then introduced, then re-introduced which makes it difficult for the readership.

Author Response

1.Thank you for your valuable feedback. We have revised the abstract to provide clearer explanations of the study's objectives and methods, and to highlight the main findings and their significance.

2.Thank you for your suggestion. We have reorganized the content in the abstract to discuss Ekman's model first, followed by the concepts of EEG and artificial intelligence.

3.Thank you for your feedback. Relevant literature on electroencephalography (EEG) and virtual reality (VR) has been incorporated into the introduction section.

4.Thank you for pointing that out. We have integrated the related work section into the introduction.

5.Thank you for your inquiry. The selection of undergraduate and graduate students as participants was based on several considerations. Firstly, these two groups typically exhibit higher academic proficiency and relatively good physical health, which helps minimize external factors' interference with the experimental results. Secondly, they represent a broad age range and diverse educational backgrounds, making the research findings more generalizable and representative. Lastly, undergraduate and graduate students usually possess strong learning and adaptation abilities, enabling better comprehension and adherence to the experimental requirements and procedures. The sample size of 32 participants was a result of balancing research design considerations and resource constraints. The choice of this sample size aimed to strike a balance between statistical power of the experiment and feasibility of resources. While 32 participants may not encompass all individual differences, such sample sizes are relatively common in similar experiments for achieving significance and reliability of results.

6.Thank you for your question.Our participants were recruited through recruitment advertisements posted on campus. These advertisements included the theme and criteria of our study, such as age range and health status. We ensured that ethical and legal standards were followed in recruiting participants, and strict recruitment procedures were adhered to.

7.Thank you for your feedback. We have taken note of the issues you pointed out and made the necessary revisions in the manuscript.

8.Thank you for your question. The reason we chose SVM as the main analysis model is due to its effectiveness in handling high-dimensional data and generalizing to unseen data. Additionally, SVM is widely used in classification tasks and has shown good performance across various fields. We will further elaborate on this choice in the revised manuscript.

9.Thank you for bringing this to our attention. We will ensure to include the appropriate reference for scikit-learn (SK-Learn) in the manuscript.

10.Thank you for your feedback. The main reason for choosing StratifiedKFold stratified cross-validation is to ensure a relatively balanced distribution of data categories when dividing the training and testing sets, thereby more accurately assessing the performance of the model.

11.Thank you for your inquiry. When selecting the kernel function, we used a grid search algorithm and set the parameter range for kernel to 'linear' and 'rbf'. Through this method, we were able to determine the optimal parameter values for SVM to achieve better classification accuracy.

12.Thank you for raising this question. We chose to use five-fold cross-validation instead of the standard ten-fold validation, considering the size of our dataset and the computational resources available for training and evaluation. While ten-fold cross-validation is a common standard, recent studies suggest that for larger datasets, five-fold cross-validation can provide comparable results at lower computational costs. However, we acknowledge that the choice of cross-validation method is an important consideration and will consider using ten-fold validation in future work.

13.Thank you for your feedback. We have made the necessary revisions to Figure 15 to ensure clarity and readability.

14.Thank you for your feedback.In our study, the presented accuracies are cross-validated accuracies. Therefore, the numbers in Table 7 reflect the averages of cross-validation, not individual sample values. Regarding the significance of accuracy results, we employed permutation tests for evaluation. We will provide more detailed explanations and statistical results in the revised manuscript to ensure the credibility and significance of the accuracy results are thoroughly explained.

15.Thank you for your feedback. We have revised the conclusion section of the manuscript.

Reviewer 4 Report

Comments and Suggestions for Authors

Differential Brain Activation for Four Emotions in VR-2D and VR-3D Modes

 I read the manuscript with interest and the authors can find my appraisal as follows: The authors stated". This advancement enables machines to perceive human emotional states, thereby making interactions more ‘empathetic’[6]" This sentence needs to be rewritten. Indeed, empathy is a complex construct and the authors need to specify in which manner this interaction is empathetic. Empathy can be emotional or cognitive. Please revise. Similarly, the authors need to be more specific about the role of psychophysiological signals in emotion recognition. EEG can detect brain signals from the scalp. However, these signals can be processed to be useful to study emotion recognition. Qualitative EEG can recognize electric alterations in the brain and it is usually used for clinical purposes. In the introduction, l read a lot of abbreviations, which can difficult the read. It is correct, but I advise to limit the use of abbreviations to facilitate the reading. Please, remove the sentence from line 96. I appreciate that the authors described the sections of the manuscript, but it can be omitted. The aims and the hypotheses must be added at the end of the introduction. Indeed, I advise avoiding section 2 and integrating it in the introduction.

The participants had no history of neuropsychiatric disorders. The authors need to add in which manner they assessed this. Moreover, 32 subjects were recruited. However, the authors did not add the estimated sample size. Please add. 

The authors used a VR+32-Channels EEG system. Did you check the presence of the interference between the 2 instruments?

 The authors used a one-way ANOVA to analyze the data, but they did not report Levene’s test.

For the SVM, the authors reported that Kernel was set on “linear”. It is not clear, please rephrase.

Despite the complexity in the methods, the results were written in a clear and intuitive way.

Tables 4 and 5: I invite the authors to use p< or P= (in that case, the second option is better) and to use bold to highlight the significant p values.

In the Discussion, I agree with the limitations, but I advise to discuss better the obtained results with previous imaging or psychophysiological studies about natural vision. 

 Please edit the English language and check the manuscript for typos. 

Author Response

1.Thank you for your detailed feedback. We appreciate your suggestions for improving the clarity and specificity of our manuscript. 

Regarding the sentence mentioning empathy, we have made modifications to provide a more specific explanation. We have also reviewed the use of abbreviations in the introduction section and limited their use to enhance readability. As per your advice, the sentence at line 96 has been removed. Additionally, we have integrated the aims and hypotheses at the end of the introduction, and merged the content of Section 2 into the introduction to simplify the manuscript's structure.

Thank you for your guidance and recommendations.

2.Thank you for your  feedback.We obtained participants' medical history information through a questionnaire survey.

3.Thank you for your  feedback. Prior to data collection, we conducted thorough signal calibration for both the VR and EEG systems to ensure optimal performance and minimize signal crossover. Furthermore, during the signal preprocessing stage, we carefully examined the EEG signals and removed any artifacts related to power line interference.

4.Thank you for the reminder. We have added the reporting of Levene's test in the manuscript.

5.In the manuscript, we set the search range for the SVM parameter "kernel" to 'linear' and 'rbf'. Through ten-fold cross-validation, we determined the optimal parameter values for SVM to achieve better classification accuracy.

6.Thank you for your suggestion. We have made revisions to the discussion section as per your advice.

7.Thank you for your feedback. We have carefully reviewed the manuscript and made necessary edits to improve the English language and ensure there are no typos.

Round 2

Reviewer 1 Report

Comments and Suggestions for Authors

1.       “Specifically, the main research contributions of this paper are summarized as follows:”. You are advised to present innovative contributions of the paper.

a.       No reaction. Innovative contributions were not presented.

2.       Recent researches have extensively explored the application of virtual reality in EEG and compared differences in brain activation between traditional 2D and VR-3D environments.

a.       “Bilgin et al. [23] designed.” Bilgin et al. [23] is not recent, year 2019;

b.       “He et al. [24] presented.” He et al. [24] is not recent, year 2018;

c.       “Kweon et al. [25].” Kweon et al. [25] is not recent, year 2018

 3.       This paper processed the preprocessed EEG signals”. Can the paper process EEG signals??

a.       No reaction. Do you still think that a paper can process EEG signals?

Comments on the Quality of English Language

Moderate editing of English language is required

Author Response

1.Thank you for your feedback. We have made the necessary revisions in the manuscript.

2.Thank you for your feedback. We have made the necessary revisions in the manuscript.

3.Thank you for your feedback. We have made the necessary revisions in the manuscript.

Reviewer 3 Report

Comments and Suggestions for Authors

This is the second review round for the manuscript : Differential Brain Activation for Four Emotions in VR-2D and VR-3D Modes. 

Please see my comments below.

Introduction:

1. Many adjectives are employed which not common to scientific publication and is usually used by ai-text generated softwares (was this used). Formal examples include: intricate (line 31), crucial (line 33), profoundly (line 34)intriguing domain (line 44), pivotal concern (line 50). I give the benefit of the doubt to the authors wether this was generated or not. The main issue is that these adjectives are charged with a valence that missleads the readership. For example, ''crucial role in many aspects'' does not explain why it is crucial.

2. While it is appreciated that the emotional models was added to the introductory paragraph, they are presented as a listing of models but there is no material that ties these models together. Line 41 mentions that this study adopts the model structure : this is a methodology and belongs to the methodological section.

3. Line 93: what is meant by application of virtual reality in EEG? Virtual reality cannot be applied to EEG. They can be used together to assess something, to do something, etc.

4. The aim of the study is much clearer in this version of the manuscript. 

5. Lines 116-124 are not contributions. Contributions have to highlight : why is your work relevant and important. From the introduction, it is still not clear what the problem is, and what is this study attempting to solve. The authors are encouraged to further develop line 110-111 to account for this and then explain why VR-2D/VR-3D coupled with EEG could help in solving this problem.

Methods:

1. While the authors provided an explanation as to why these participants were selected, the manuscript has not changed in regards to this information. To the readership, participants selection and recruitment is still opaque.

2. My previous comment on the metrics measurements selection process (validity and fidelity scores) has not been adressed.

3. Line 211-215 should be in the participants section as it is mandatory that participants consent to participate in the study (see Declaration of Helsinski).

4. Choice of model (SVM) is still undefined in the manuscript and relevant decision for choosing the model has not been added to the manuscript.

5. In this revised version, the methodology is still overly complex and none of the material was exported to supplementary material to ease the readership. Rationale for choosing the aforementioned technology is not discussed. The methodology therefore remains opaque to the readership. The authors responded to my previous comment on SVM ''The reason we chose SVM as the main analysis model is due to its effectiveness in handling high-dimensional data and generalizing to unseen data. Additionally, SVM is widely used in classification tasks and has shown good performance across various fields.''. This is not a reason as handling high-dimensional data is true for most of any machine learning models these days. The choice of model is very important as they all come with pros and cons. The manuscript is lacking on formal explanations on the choice of model and limitations of the model.

6. Cross-validation: the authors provided the following explanation in their response (but the manuscript was not updated accordingly): While ten-fold cross-validation is a common standard, recent studies suggest that for larger datasets, five-fold cross-validation can provide comparable results at lower computational costs. Please mention these recent studies and cite them as this is very uncommon for the field to use a five-fold cross validation. 

7. While reading the manuscript, it is still very unclear to me why low-level model such as SVM is used, and then processing deep learning models are used which renders the overall process very complex with a lot of potential biases and limited external validity. This is why it is very important to highlight the rational behind each model choices.

Results:

1. Most of the figures are still unreadable at this stage. This has to do with resolution and image size.

2. Clinical and statistical significancy should be part of the methodology and not the result as choosing as p-value of less than 0.05 is not a result (line 445-446)

3. Analysis results such as F1 score are presented but where yet never introduced in the methodology. There is a missmatch between the methods and the reported results that should be adressed.

Discussion:

1. My comment on including the limitations regarding the models was not adressed.

Minor comment:

- The verb tense varies across the manuscript. This should be made uniform across the manuscript.

Comments on the Quality of English Language

The verb tense varies across the manuscript. This should be made uniform across the manuscript.

Author Response

1.Thank you for your thoughtful feedback on the language used in our manuscript. We acknowledge your concerns regarding the use of adjectives that may not align with typical standards of scientific writing. We have carefully reviewed and revised the manuscript to ensure that the language accurately reflects the scientific content.

2.Thank you for your feedback. We have made the necessary revisions in the manuscript.

3.Thank you for your feedback. We have made the necessary revisions in the manuscript.

4.Thank you for your affirmation.

5.Thank you for your feedback. We have made the necessary revisions in the manuscript.

6.Thank you for your feedback. The information regarding participant selection and recruitment has been corrected in the manuscript.

7.Thank you for your feedback. The SAM validity and fidelity psychometrics propriety has been discussed in the manuscript.

8.Thank you for your feedback. The section previously located in lines 211-215 of the manuscript has been moved to the Participants and Ethics section.

9.Thank you for your feedback. We have defined and referenced SVM in the manuscript.

10.Thank you for your feedback. Because our collected dataset has its own characteristics, SVM is not biased towards VR-2D or VR-3D data. Additionally, SVM is widely used in classification tasks and has shown good performance across various domains. Therefore, we chose the SVM-based method as the initial classification approach. Subsequently, to further improve the accuracy of EEG emotion recognition, we introduced deep learning methods.

11.Thank you for your feedback. We have revised the five-fold cross-validation to ten-fold cross-validation in the manuscript.

12.Thank you for your feedback. We have adjusted the clarity of the images in the manuscript.

13.Thank you for your feedback. We consider this part to be one of the results of the statistical analysis.

14.Thank you for your feedback. We have added an introduction to evaluation metrics such as F1 score in the section "Machine Learning-Based Classification of Brain Activation States".

15.Thank you for your feedback. We have discussed the limitations of the model in the "Future Work" section of the manuscript.

Reviewer 4 Report

Comments and Suggestions for Authors

The authors addressed my concerns and I found the manuscript improved. 

Author Response

Thank you for your time and effort in reviewing our work. Thank you.